# Non-Invasive versus Invasive Samples for Zika Virus Surveillance: A Comparative Study in New Caledonia and French Guiana in 2015–2016

**DOI:** 10.3390/microorganisms9061312

**Published:** 2021-06-16

**Authors:** Marie-Alice Fraiture, Wim Coucke, Morgane Pol, Dominique Rousset, Ann-Claire Gourinat, Antoine Biron, Sylvia Broeders, Els Vandermassen, Myrielle Dupont-Rouzeyrol, Nancy H. C. Roosens

**Affiliations:** 1Transversal & Applied Genomics (TAG), Sciensano, rue Juliette Wytsman 14, 1050 Brussels, Belgium; marie-alice.fraiture@sciensano.be (M.-A.F.); sylvia.broeders@sciensano.be (S.B.); els.vandermassen@sciensano.be (E.V.); 2Quality of Laboratories, Sciensano, rue Juliette Wytsman 14, 1050 Brussels, Belgium; wim.coucke@sciensano.be; 3URE Dengue et Arboviroses, Institut Pasteur of New Caledonia, 11 avenue Paul Doumer, BP 61, CEDEX, 98845 Noumea, New Caledonia; mpol@pasteur.nc (M.P.); ann-claire.gourinat@cht.nc (A.-C.G.); antoine.biron@cht.nc (A.B.); mdupont@pasteur.nc (M.D.-R.); 4Laboratoire de Virologie CNR Arbovirus, Institut Pasteur de la Guyane, 23 avenue Pasteur, BP 6010, CEDEX, 97306 Cayenne, French Guiana; drousset@pasteur-cayenne.fr

**Keywords:** Zika virus, diagnostic, RT-qPCR, serum, urine and saliva samples, public health concern

## Abstract

Zika virus, an arbovirus responsible for major outbreaks, can cause serious health issues, such as neurological diseases. In the present study, different types of samples (serum, saliva, and urine), collected in 2015–2016 in New Caledonia and French Guiana from 53 patients presenting symptoms and clinical signs triggered by arbovirus infections, were analyzed using a recently developed, and in-house validated, 4-plex RT-qPCR TaqMan method for simultaneous detection and discrimination of the Zika and Chikungunya viruses. Subsequently, statistical analyses were performed in order to potentially establish recommendations regarding the choice of samples type to use for an efficient and early stage Zika infection diagnosis. On this basis, the use of only urine samples presented the highest probability to detect viral RNA from Zika virus. Moreover, such a probability was improved using both urine and saliva samples. Consequently, the added value of non-invasive samples, associated with a higher acceptance level for collection among patients, instead of serum samples, for the detection of Zika infections was illustrated.

## 1. Introduction

The massive concentration of humans and animals in one place, combined with the high exchange of persons, animals, and products worldwide, drastically increases the risk of exposure to new, emerging, and exotic infections with an enlarged potential for spread [1,2,3,4,5,6,7]. Among the emerging pathogens, Zika virus (ZIKV), a single-strand RNA flavivirus, has been responsible for several outbreaks, such as that in 2007 in the Federated States of Micronesia; in 2013–2014 in New Caledonia and other Pacific islands; and in 2014–2016 in Central and Latin America and the United States. Moreover, several cases of travelers returning from areas with endemic/epidemic Zika fever have been reported [1,2,3,4,5,6,7]. This arbovirus is mainly transmitted to human by *Aedes aegypti* and *Aedes albopictus* mosquito bites, of which geographical expansion is accentuated by urbanization, global warming, and travel [1,6,8]. ZIKV can also be transmitted through body fluids during sexual relations, as well as from mother to fetus via the placenta. In infected humans, ZIKV presence was reported, for example, in blood, saliva, urine, semen, breast milk, nasopharyngeal secretion, cerebrospinal fluid, brain tissue, umbilical cord, amniotic fluid, and placenta [1,2,6,8,9,10,11]. The majority of infected patients present no specific clinical features. Indeed, only around 20–25% of them have symptoms and clinical signs, such as headache, fever, rash, conjunctivitis, myalgia, and arthralgia/arthritis, which are usually resolved in 7 days. However, even if the symptoms and clinical signs are generally benign, such infections can cause serious health issues, including fetus malformations, brain defects in newborns, pregnancy problems, Guillain-Barré syndrome, and other neurological and autoimmune complications [1,6,8,9,10,11]. Consequently, the World Health Organization (WHO) raised a “Public Health Emergency of International Concern” regarding this emerging pathogen [1,2,3,4].

The clinical diagnosis of ZIKV infections is challenging due to frequent asymptomatic infections, as well as the similarity of the clinical signs and symptoms with other arbovirus infections, such as Dengue virus and Chikungunya virus (CHIKV). To determine ZIKV infection, laboratory analyses therefore need to be performed. Among the laboratory techniques, serological and molecular approaches exist and present both advantages and drawbacks. Serological approaches have widely been used; however, such techniques are less sensitive for early stage diagnosis (first days after infection), which is essential to quickly identify the causative pathogen of an emerging epidemic as it markedly increases the chance of success for any countermeasures to constrain the disease. Therefore, to overcome these crucial issues, molecular approaches targeting nucleic acids of interest, such as the reverse transcription real-time polymerase chain reaction (RT-qPCR), are generally privileged [4,8,12,13,14]. These analyses are usually performed by diagnostic laboratories using serum samples; however, the collection of serum samples does not represent the simplest strategy for massive implementation because it requires medical infrastructure and healthcare teams. In addition, the collection of blood samples can be difficult for application for an entire population, such as with neonates, young children, the elderly, hemophiliacs, and refractory patients, especially if their symptoms and clinical signs are moderate. In this context, the possible use of non-invasive samples (i.e., urine and saliva) presenting a higher acceptance level for collection has recently been investigated. On this basis, the detection of Zika viral RNA in such non-invasive samples from infected patients was successfully demonstrated. In addition, the use of urine samples offered a broader detection window compared to the use of saliva and serum samples [2,4,8,9,14,15,16,17,18,19,20,21,22,23,24,25,26].

In the present study, the potential added-value of non-invasive samples (urine and saliva) in comparison to classical serum samples was assessed for ZIKV in terms of early diagnostics and survey. To this end, a large set of urine, saliva, and serum samples was collected in 2015–2016 in New Caledonia and French Guiana from 53 patients presenting symptoms and clinical signs associated with arbovirus infections. All the samples were tested using a recently developed and in-house validated 4-plex RT-qPCR TaqMan method for simultaneously detecting and discriminating ZIKV and CHIKV, which often co-circulate in the same geographical areas and for which the number of co-infection cases is more and more reported [14,27,28,29,30]. The generated results were then analyzed statistically in order to highlight the strengths and weaknesses of the tested samples to support diagnostic laboratories.

## 2. Materials and Methods

### 2.1. Sample Collection and RNA Extraction

The presented study, including publication of data related to arbovirus infection, was approved by the Comité Consultatif d’Ethique de Nouvelle-Calédonie (February 2015) and by the Comité de Protection des Personnes Ile de France I (March 2015-13851 and January 2018-14793). Human biological samples used in this study were obtained from patients after written informed consent. All samples were collected based on an ethical agreement (n° 2015-mars-13851 and January 2018-14793). Serum, urine, and saliva samples were collected in 2015–2016 from 47 patients from New Caledonia and 6 patients from French Guiana. These patients presented symptoms and clinical signs associated with arbovirus infections, including ZIKV (Table 1 and Appendix A). For thirteen patients (NC-19, NC-23, NC-26, NC-37, NC-40, NC-41, NC-44, NC-46, NC-47, GY-1, GY-4, GY-6, and GY-8), a second sample test was carried out (Table 1 and Appendix A). For each patient number, the A and B labels are, respectively, associated to indicate the first and second sample tests. The number of days since onset of symptoms and clinical signs was recorded for each sample. For serum, 1 mL was sampled from whole blood (collected in a dry tube) centrifuged at 3000 rpm for 5 min. For urine, 20 mL were sampled in a sterile vessel and centrifuged at 3000 rpm for 10 min. The supernatant was discarded, except for 1 mL to resuspend the pellet. For saliva, 1 mL was collected using the ORACOL Saliva Collection Device (Malvern Medical Developments, Worcestershire, UK), as previously described [31]. All samples were stored at −80 °C.

RNA was extracted using the MagNA Pure LC Total Nucleic Acid Isolation Kit (Roche, Indianapolis, IN, USA), for samples from New Caledonia patients, or the QiAamp Viral RNA mini Kit (Qiagen, Kista, Sweden), for samples from French Guiana patients, according to the manufacturer’s instructions. RNA extraction was performed in 2016–2017.

Reference RNA for ZIKV (AMPLIRUN^®^ ZIKA VIRUS RNA CONTROL, VIRCELL^®^) and CHIKV (AMPLIRUN^®^ CHIKUNGUNYA VIRUS RNA CONTROL, VIRCELL^®^) were obtained from Labconsult (Schaerbeek, Belgium).

### 2.2. One Step RT-qPCR

All RT-qPCR assays were carried out on a CFX96 Touch^TM^ (BIO-RAD, Hercules, CA, USA) system using the SuperScript^®^ III Platinum^®^ One-Step RT-qPCR Kit (Invitrogen, Waltham, MA, USA) according to the manufacturers’ instructions. All samples were tested using a 4-plex RT-qPCR targeting both CHIKV, via the CHIKV (a) and CHIKV (b) methods, and ZIKV, via the ZIKV (a) and ZIKV (b) methods (Table 1 and Appendix A) [14]. This 4-plex RT-qPCR, previously developed and validated by Broeders et al. (2020), uses two independent target sequences for each virus, improving the reliability of the assay, such as in the case of mutations in one of the target sequences. This 4-plex was also assessed as specific and sensitive (as low as 5–100 cp) [14]. One reference RNA for CHIKV and one for ZIKV were used as positive control [14]. In each PCR reaction, 5 µL of RNA that was extracted from symptomatic patient samples were tested. Each PCR assay was tested in duplicated and the average Cq is indicated in Table 1. The RT-qPCR program consisted of the following successive steps: (i) cDNA synthesis for 15 min at 50 °C; (ii) initial DNA polymerase activation for 2 min at 95 °C; (iii) and 45 amplification cycles for 15 s at 95 °C (denaturation) and for 30 s at 60 °C (annealing and extension). All reactions were set up on ice and water was used as the no-template control [14]. All signals with a Cq value greater than 38 were considered negative. RT-qPCR assays were performed in 2017–2018.

### 2.3. Statistical Analysis

#### 2.3.1. Probability of Detection

Based on the Cq values indicated in Table 1, the number of positive and negative samples per kind of sample (serum, urine, and saliva) was used to determine the probability of detecting ZIKV (Figure 1 and Appendix A). The significance of this probability using each kind of sample is represented via the p-value (Figure 1B and Appendix A). This analysis was performed using S-plus (version 8.0, TIBCO Software Inc, Carlsbad, CA, USA) for Linux, a generalized linear model for binary data. A logit link was fit using the method as a fixed factor and the patient as a random factor. The comparison between the two ZIKV methods were corrected for simultaneous hypothesis testing, according to Sidak.

#### 2.3.2. Signal Precocity

Based on the data in Table 1, the Cq values were compared between samples (urine, saliva, and serum) per patient when both compared samples presented a positive signal (Cq value ≤ 38). For each compared sample, the Cq value from the ZIKV (a) method, indicated in Table 1, was taken into account to calculate the Cq value difference on which the *p*-value was determined (Table 2). This analysis was performed using a Studentized t-test followed by a correction for simultaneous hypothesis testing, according to Sidak. On this basis, the significance of the signal precocity between the compared samples was assessed (Table 2).

## 3. Results and Discussions

Serum, urine, and saliva samples were collected in 2015–2016 in New Caledonia and French Guiana from 53 patients presenting symptoms and clinical signs associated with arbovirus infections, including mainly fever, myalgia, arthralgia, cephalgia, nausea, retro-orbital pain, and skin rash (Appendix A). This large set of 193 samples was analyzed for the potential presence of viral RNA of CHIKV and ZIKV. To this end, a 4-plex RT-qPCR targeting both CHIKV, via the CHIKV (a) and CHIKV (b) methods, and ZIKV, via the ZIKV (a) and ZIKV (b) methods, previously developed and assessed as specific and sensitive by Broeders et al. (2020), was applied (Table 1 and Appendix A) [14]. Given that this 4-plex RT-qPCR included two methods for each targeted virus, the detection of these viruses was confirmed when at least one of the corresponding method presented a positive signal. Moreover, a patient was considered as infected when at least a positive signal was observed for either in the serum, urine, or saliva samples.

Among the 53 tested patients, none of them presented viral RNA from CHIKV while viral RNA from ZIKV was observed in 22 patients (Table 1). Such observations are consistent with the fact that, in contrast to CHIKV, ZIKV was one of the most circulating arboviruses during the sampling period [32,33]. For the 22 patients infected with ZIKV, 10 serum, 22 urine, and 18 saliva samples showed a positive signal for viral RNA from ZIKV, suggesting the potential capacity of non-invasive samples to cover a larger number of infected cases than invasive samples. In order to investigate such a hypothesis, statistical tests were performed using the generated RT-qPCR results (Figure 1 and Appendix A). On this basis, the relevance of using serum, urine, and/or saliva samples was assessed as follow.

First, the probability of detecting viral RNA from ZIKV was determined for each individual type of sample used (serum, urine, or saliva), regardless the day of declaration of symptoms and clinical signs (Figure 1A). The urine samples presented the highest probability of detection in comparison to the serum and saliva samples. This probability was calculated as being significantly higher for the urine samples than for the serum samples (Figure 1B). Interestingly, the exclusive use of serum samples, which is commonly done in arbovirus diagnostic laboratories [8,17], gave the lowest probability of detection. On this basis, the use of urine samples to diagnose ZIKV, which represented the most appropriate individual test, is strongly advised. Such results are supported by recent studies in which similar recommendations were also made [4,9,17,21,22,23,34,35,36].

Second, the probability of detecting viral RNA from ZIKV was assessed using combinations of different types of samples (serum, urine, and/or saliva), regardless of the number of days of declaration of symptoms and clinical signs (Figure 1A). In comparison with the individual use of each type of sample (serum, urine, or saliva), the probability of detection was increased by combining the analyses of the urine and saliva samples, highlighting the added value of non-invasive samples in terms of detection. The probability of detection associated with the combination of urine and saliva samples was also determined as significantly higher than using only the serum or saliva samples (Figure 1B). The use of both the urine and saliva samples, which represented the most appropriate combined test, is thus strongly advised.

Finally, the potential benefit of using certain types of samples to detect ZIKV infections at an early stage was investigated. To this end, the signal precocity, based on the observed Cq values indicated in Table 1, for each type of sample (serum, urine, or saliva) used individually, was analyzed (Table 2). On this basis, in comparison with the serum samples, the saliva samples presented a positive signal and ZIKV infection was significantly more precocious. However, between the urine and saliva samples as well as between the urine and serum samples, no significant difference was observed in terms of signal precocity.

According to all these results, the present comparative study supports the use of non-invasive samples to detect ZIKV infections. Indeed, compared to serum samples, non-invasive samples offered a higher probability of detection and the saliva samples presented a particular item of interest to target such infection at an early stage. As a perspective, the use of a larger number of samples and patients could be interesting in order to support such data.

## 4. Conclusions

Based on the present study, the use of one or a combination of non-invasive samples instead of serum samples is highly recommended for the detection of viral RNA from ZIKV at early stage. Indeed, the use of urine samples, as well as the use of both urine and saliva samples, were highlighted as being the most appropriate tests in this study. In addition to their better acceptance for collection among patients, non-invasive samples can easily be collected without any medical background. This can be important in the case of outbreaks, as this would allow supporting overworked healthcare teams. Non-invasive samples are also ideal for either at-home collection kits or self-testing kits, as can be currently observed in the context of the COVID-19 crisis. Given the obvious added value of using non-invasive samples in diagnostics, at the logistic level, as well as in the improvement of diagnostic accuracy, the comparison approach proposed in this study could systematically be applied for all viral infections (e.g., Dengue virus). On this basis, the type(s) of samples being the best compromise for an efficient diagnostic and outbreak surveillance could be determined.

## Figures and Tables

**Figure 1 microorganisms-09-01312-f001:**
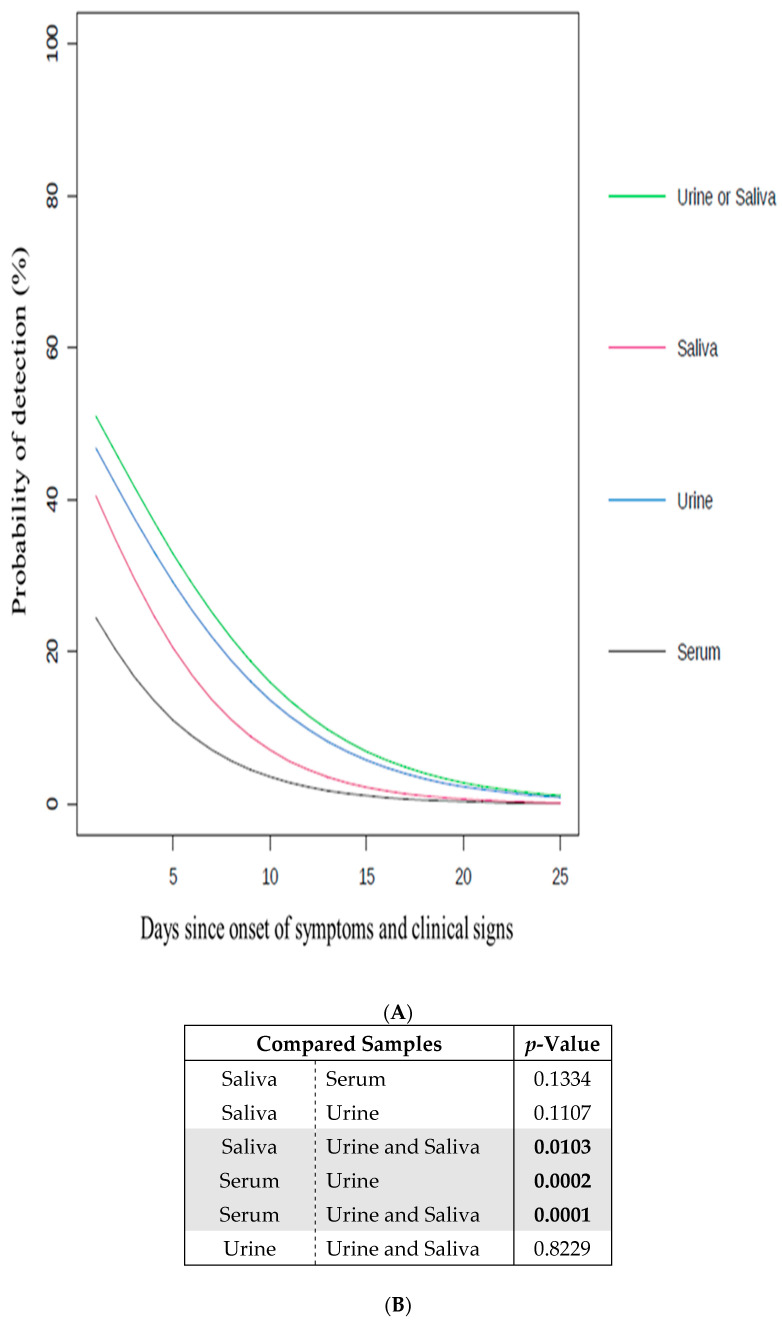
Impact of the samples on the probability to detect ZIKV. (**A**) Probability of detection (%) of ZIKV in serum, urine and/or saliva samples according to days since onset of symptoms and clinical signs (going from 1 to 25) (Appendix A). Serum (black line), urine (blue line), saliva (red line), and urine and/or saliva samples (green line); (**B**) the significance of the difference between the curves in (**A**) is indicated by the *p*-value. The *p*-values in bold are considered as significant (<0.05). The significant sample comparisons are highlighted in grey.

**Table 1 microorganisms-09-01312-t001:** RT-qPCR diagnostic results from patients presenting symptoms and clinical signs associated with arbovirus infections.

Patient Number	Days Since Onset of Symptoms/Clinical Signs	SERUM		URINE		SALIVA	
		ZIKV (a)	ZIKV (b)	ZIKV (a)	ZIKV (b)	ZIKV (a)	ZIKV (b)
NC-1A	3	34.9	37.4	-	-	31.4	32.8
NC-2A	4	37.6	-	-	-	-	-
NC-3A	4	35.7	-	31.4	35.1	28.4	29.8
NC-4A	3	-	-	30.7	33.4	32.6	32.6
NC-8A	3	-	-	25.2	28.3	-	-
NC-9A	3	35.7	-	33.0	36.2	30.9	32.3
NC-10A	5	-	-	23.8	26	37.6	-
NC-11A	5	-	-	34.1	36.6	-	-
NC-12A	1	35.9	-	25.8	28.9	29.8	31.1
NC-13A	2	33.8	37	34.4	35.8	31.3	34.3
NC-14A	2	-	-	28.2	29.2	30.7	31.7
NC-15A	4	-	-	32.0	33.7	37.6	-
NC-17A	2	-	-	32.8	35.6	36.5	-
NC-19A	1	-	-	29.7	30.6	28.3	29.4
NC-19B	2	-	-	30.0	31.40	32.2	33.8
NC-21A	1	31.2	34.7	37.2	-	29	31.4
NC-23A	2	-	-	31.0	33.9	30.4	32.3
NC-23B	6	-	-	30.9	32.9	37.7	-
NC-26A	2	35.8	-	37.0	-	38	-
NC-26B	6	37.4	-	37.9	-	-	-
GY-1A	2	33.9	34.2	-	-	-	-
GY-1B	3	-	-	32.0	32.4	36.7	37.6
GY-4A	3	-	-	37.8	37.1	-	-
GY-5A	4	-	-	-	-	32.2	33.2
GY-6A	1	-	-	31.3	31.9	-	-
GY-8A	6	-	-	34.2	34.5	-	-

For each patient, the serum, urine, and saliva samples were tested using a 4-plex targeting CHIV, via the CHIKV (**a**) and CHIKV (**b**) methods, and ZIKV, via the ZIKV (**a**) and ZIKV (**b**) methods. Only results associated with patients with a positive signal are presented. For each sample, the corresponding number of days since onset of symptoms and clinical signs is indicated. For each patient number, the A and B labels are associated to respectively indicate the first sample test and the second sample test. All Cq values above 38 are considered as negative (not detected) and are represented by the symbol “-”. Each Cq value is the average of a duplicate (standard deviation means of 0.5). In the patient number, the symbols NC and GY, corresponding, respectively, to New Caledonia and French Guiana, indicate the sampling areas.

**Table 2 microorganisms-09-01312-t002:** Difference in the RT-qPCR signals observed between the serum, urine, and saliva samples. Cq value difference is based on the Cq value from the ZIKV (a) method indicated in Table 1 for each compared sample that were both presenting a positive signal (Cq value ≤ 38). *p*-value in bold is considered significant (<0.05).

Compared Samples	Cq Value Difference	*p*-Value
Serum-Saliva	−4.4	**0.0095**
Serum-Urine	−1.3	0.8998
Saliva-Urine	−1.8	0.4974

## Data Availability

Not applicable.

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
