# Peer review of "Non-Invasive versus Invasive Samples for Zika Virus Surveillance: A Comparative Study in New Caledonia and French Guiana in 2015–2016"

_microorganisms, 2021, doi:10.3390/microorganisms9061312_

Round 1

Reviewer 1 Report

In this study, non-invasive sampling of patients was compared with serum sampling to detect Zika virus RNA. The possibility to use non-invasive sampling is much more convenient and this study also showed that it can be more sensitive. The purpose and results of the study is therefore of interest, however I have a number of questions and suggestions, see below. Overall, the scientific language needs to be improved significantly.

Introduction

The introduction needs to be less wordy and focus more on details on how ZIKV is excreted (what samples can potentially be taken) For example P2L75-76: provide some more detailed information from what these studies showed.

Why do you include Chikungunya and not Dengue virus?

Material and Methods

What genes are the target for your primers and probes?

Explain the difference between method a and b and why you use two targets per virus?

What is the PCR sensitivity and efficiency for the different targets?

Add that all PCR’s were performed in duplicates.

Results and discussion

Do you find it surprising that none were CHIV positive as it’s common in the region? Why do you think they were all negative?

Do you have the possibility to do confirming serology? For example, it would be interesting to confirm NC-2A which is only borderline positive for one sample, one target.

I think S2 should be in the paper, but edited so it’s easier to read, for example including total number instead of neg and pos in different columns.

What were the Ct differences within replicates? Were they always significantly smaller that the differences reported in Fig 1?

Table 1

Remove CHIV from the table to make it easier to read. It’s enough to write in the text that all were negative for CHIV.

Make it easier to interpret which second samples came from the same patient. The separating lines/coding now seems inconsistent (e.g. are GY-4A and GY-5A from the same patient? And same for GY-6A and GY-8A.).

Remove the details below the table of patient codes that were negative.

Figure 1

Use different line formatting instead of different colours (or use both).

Add the points that the lines are computed from.

Remove boxing gloves and “versus”.

Author Response

Reply to the reviewers’ comments

Reviewer #1

In this study, non-invasive sampling of patients was compared with serum sampling to detect Zika virus RNA. The possibility to use non-invasive sampling is much more convenient and this study also showed that it can be more sensitive. The purpose and results of the study is therefore of interest, however I have a number of questions and suggestions, see below.

Thank you.

Overall, the scientific language needs to be improved significantly.

English language was revised.

Introduction: The introduction needs to be less wordy and focus more on details on how ZIKV is excreted (what samples can potentially be taken)

The introduction was reworked and more details about Zika virus presence in infected humans was added.

For example P2L75-76: provide some more detailed information from what these studies showed.

These studies indicated also that the use of urine samples offered a broader detection window compared to the use of saliva and serum samples. This point was added in the introduction.

Introduction: Why do you include Chikungunya and not Dengue virus?

The authors acknowledge the concern for DENV and the importance of having also good detection and discrimination of DENV seen its co-occurrence with CHIKV and ZIKV and the similarity in symptoms. Here, we used a developed 4-plex targeting ZIKV and CHIKV. The same study could be performed using appropriated real-time PCR methods targeting any viruses of interest, such as DENV (see conclusion).  

Material and Methods: What genes are the target for your primers and probes? Explain the difference between method a and b and why you use two targets per virus?

The names of the targeted genes for each method were added in Table S2. In addition, as explained in Broeders et al (2020), two targets per virus were selected to enlarge the coverage of the multiplex method (“The use of two independent target sequences for each virus enhances the reliability of the assay, as false-negative results for the first method due to the occurrence of mutations can be covered by the second method”). This point was also added in section 2.2.

Material and Methods: What is the PCR sensitivity and efficiency for the different targets?

The 4-plex real-time PCR was not developed and validated in this study. Such information are detailed in Broeders et al. (2020), showing that the 4-plex performance was assessed as specific and sensitive (as low as 5-100 copies of the targets). This point was added in section 2.2.

Material and Methods: Add that all PCR’s were performed in duplicates.

This was added.

Results and discussion: Do you find it surprising that none were CHIV positive as it’s common in the region? Why do you think they were all negative?

Most of samples used in these study were collected in New Caledonia between 2015-2016. During this period, DENV and ZIKV were the main circulating arboviruses. Indeed, as referenced by Inizan et al (2019) and Roth et al (2014), the main CHIKV outbreak occurred in the Pacific region in 2013-2014. During this period, less than 50 CHIKV cases were detected in New Caledonia and most of them were imported cases from French Polynesia. Based on this epidemiological considerations, no CHIKV detection was thus not surprising. This point was mentioned in page 5.

Results and discussion: Do you have the possibility to do confirming serology? For example, it would be interesting to confirm NC-2A which is only borderline positive for one sample, one target.

We thank the reviewer for this remark. Running a serology test would have been interesting if we had a late serum sample in order to highlight a seroconversion or at least an increase of IgM between samples. Unfortunately, we do not have this late serum sample and thus we do not have the possibility to run such test.

Results and discussion: I think S2 should be in the paper, but edited so it’s easier to read, for example including total number instead of neg and pos in different columns.

This table (Table S3 in the revised manuscript) was edited as suggested. For such data (intermediate analysis), the main results are illustrated in Figure 1. Therefore, this table, where associated data are detailed, was kept in the supplementary data section. However, we will adapted it if required by the editor.

Results and discussion: What were the Ct differences within replicates? Were they always significantly smaller that the differences reported in Fig 1?

The standard deviation mean (0.5) for duplicate was added in the legend of Table 1.

Table 1: Remove CHIV from the table to make it easier to read. It’s enough to write in the text that all were negative for CHIV.

As suggested, the results for the CHIKV(a) and CHIKV(b) methods were removed from Table 1.

Table 1: Make it easier to interpret which second samples came from the same patient. The separating lines/coding now seems inconsistent (e.g. are GY-4A and GY-5A from the same patient? And same for GY-6A and GY-8A.).

The separating lines were corrected.

Table 1: Remove the details below the table of patient codes that were negative.

This part was deleted.

Figure 1 : Use different line formatting instead of different colours (or use both).

Figure 1 was not modified as suggested because we believe that the different colours are relevant to illustrate the results. However, we will adapted it if modifications are required by the editor.

Figure 1 : Add the points that the lines are computed from.

The lines are associated to the days since onset of symptoms and clinical signs onset (going from 1 to 25). This point was added in the legend.

Figure 1 : Remove boxing gloves and “versus”.

The figure was adapted.

Reviewer 2 Report

Dear authors

Please see the comments below:

1-Please describe the patient’s characteristics more thoroughly, a table would be great to introduce the symptoms and why you chose them

2-Please rearrange the table 1 into a more comprehensive table, less footnote would help the readers, the tables should be clear and concise, all abbreviations must be footnoted

3-please provide sensitivity and specificity as better figures to evaluate each test

4-comparing should be depicted using bars and p values; we cannot see which test is superior based on your figures.

5-Please rewrite your discussion based on the new suggested results

6-Conclusion should be the main contributions of your work and avoid referencing in this section

Author Response

Reply to the reviewers’ comments

Reviewer #2

1-Please describe the patient’s characteristics more thoroughly, a table would be great to introduce the symptoms and why you chose them

Table S1, indicating symptoms and clinical signs from patients, was added.

2-Please rearrange the table 1 into a more comprehensive table, less footnote would help the readers, the tables should be clear and concise, all abbreviations must be footnoted

Table 1 was adapted to be more comprehensive.

3-please provide sensitivity and specificity as better figures to evaluate each test

The 4-plex real-time PCR was not developed and validated in this study. Such information are detailed in Broeders et al. (2020), showing that the 4-plex performance was assessed as specific and sensitive (as low as 5-100 copies of the targets). This point was added in section 2.2.

4-comparing should be depicted using bars and p values; we cannot see which test is superior based on your figures.

Such analysis is illustrated in Figure1, which is detailed in two parts (A and B) in order to improve the clarity of the results. The part A shows the probability of detection using a type of sample(s). The part B (where p-values are indicated) presents the significant difference(s) between all these probabilities (indicated in part A). On these basis, the use of urine as well as the use of both urine and saliva were highlighted as being the superior tests. To avoid any misunderstanding, these superior tests were highlighted in grey in the part B and an additional sentence was added in the legend.

5-Please rewrite your discussion based on the new suggested results

The results illustrated in Figure 1 are discussed in pages 5-6. In addition to the modifications added for the previous comments, the advised used of both urine and saliva was clearly specified.   

6-Conclusion should be the main contributions of your work and avoid referencing in this section

The reference was deleted. The main contribution of our work is clearly indicated in the conclusion: “ the use of one or a combination of non-invasive samples instead of serum samples is highly recommended for the detection of viral RNA from ZIKV at early stage.   

Reviewer 3 Report

In this manuscript the authors utilize a 4-plex RT-qPCR TaqMan method to detect CHIKV and ZIKV infection in various samples types. They identify the non-invasive samples of urine and saliva to be better sites of detection than the standard serum samples. This study provides important insight into the importance of which samples are being used for diagnostics. 

Specific Comments:

The authors need to more clearly explain what the difference between a and b methods are in the  RT-qPCR section. Also the RNA extraction methods used need to be included. 

Author Response

Reply to the reviewers’ comments

Reviewer #3

In this manuscript the authors utilize a 4-plex RT-qPCR TaqMan method to detect CHIKV and ZIKV infection in various samples types. They identify the non-invasive samples of urine and saliva to be better sites of detection than the standard serum samples. This study provides important insight into the importance of which samples are being used for diagnostics.

Thank you

The authors need to more clearly explain what the difference between a and b methods are in the  RT-qPCR section.

The quadruplex real-time PCR was not developed in this study. Such information are detailed in Broeders et al., 2020. This point was clarified in section 2.2 and section 3. Moreover, the names of the targeted genes by each methods were added in Table S2.

Also the RNA extraction methods used need to be included.

The RNA extraction is described in section 2.1.

Reviewer 4 Report

Dear authors,

Following my comments for the manusctipt Non-invasive versus invasive samples for Zika virus surveillance: a comparative study:

Line 35 – Please, include a reference for the first sentence here.

Line 35 – I suggest describing the taxonomic acronym ZIKV here, as Zika virus (ZIKV), and then use the acronym ZIKV elsewhere in the manuscript.

Line 35 – If describing the taxonomic entity genus Flavivirus, it should be in italics: Flavivirus. If only referring to the group of flaviviruses, use no italics and no capital letter: flavivirus.

Line 35 – This sentence is too long with five lines. I suggest dividing it in two or three sentences or if keeping as one sentence reduce the amount of details to describe ZIKV occurrence worldwide.

“Among the emerging pathogens, Zika virus, a single-strand RNA Flavivirus has been responsible of several outbreaks, such as in 2007 in Yap, Federated States of Micronesia, Pacific region, in 2013-2014 in French Polynesia, Cook Islands, Vanuatu, Salomon Islands, 37 Ester Islands and New Caledonia and other Pacific Islands, and in 2014-2016 in Central 38 America, Latin America, the Caribbean, the United States and the Pacific region.”

Suggestion:

“Among the emerging pathogens, Zika virus (ZIKV), a single-strand RNA flavivirus has been responsible of several outbreaks, such as in 2007 in the Federated States of Micronesia, in 2013-2014 in New Caledonia and other Pacific Islands, and in 2014-2016 in Latin America and the United States.”

Line 43 – Please, add a reference after “traveling”

Line 43 – Please, from now on use the acronym ZIKV for all “Zika virus”

“The Zika virus can then also be passed from infected humans to other humans through body fluids…”

Suggestion:

“ZIKV can also be transmitted through body fluids…”

Line 46 – Replace “have symptoms” by “have symptoms and clinical signs”. Fever, rash and conjunctivitis are not symptoms. Change that elsewhere in the manuscript.

Line 54 – Replace “medical” by “clinical”

Line 57 – Replace “further” by “laboratorial”

Line 59 – Replace “ineffective for the early stage diagnosis, which is essential for the treatment of individual patients and, even more, for the population in terms of public health.” by “less sensitive for the very early stage of infection.”

Line 61 – Delete “or pandemic”

Line 63 – “However, such techniques are ineffective for the early stage diagnosis, which is essential for the treatment of individual patients and, even more, for the population in terms of public health. Indeed, the ability to quickly identify the causative pathogen of an emerging epidemic or pandemic markedly increases the chances of success for any countermeasure to constrain the disease. Moreover, serological techniques present usually a lack of specificity due to the possible cross reactivity with antibodies from infections associated to other arboviruses and, even, other pathogens. Therefore, to overcome these crucial issues, molecular approaches targeting nucleic acids of interest, such as the reverse transcription real-time polymerase chain reaction (RT-qPCR), are generally privileged.”

Laboratorial methods designed to detect antibodies are the most valuable diagnostic tools for arbovirus diagnosis. There are around 500 arboviruses officially reported to the International Committee on Taxonomy of Viruses worldwide and the large majority of them is kept in enzootic cycles of transmission, which humans are accidental hosts for not mounting viremia high and long enough to act as a source of infection to arthropods. Very few exceptions including zika, dengue, chikungunya, and yellow fever, partially, have humans as amplifying hosts and only for these arboviruses the use of direct laboratorial diagnosis as virus isolation and RT-PCR is likely to be successful. It is noteworthy that the direct diagnostic is more sensitive only for a very short period of time and the success of the diagnostic is dependent on the sample storage conditions and viremia duration, while antibodies that are proteins, are more resistant than RNA and can be detected for a longer time after exposure. Indeed, the cross-reactivity observed among flaviviruses influences serological diagnostic results and the interpretation of serological tests is complex requiring careful evaluation. However, there are serological methods with high specificity as the plaque reduction neutralization test (PRNT), which is able to differentiate flavivirus infections in convalescent serum samples. Two or more flaviviruses can be distinct from each other by quantitative serological criteria. Thus, each laboratorial method used in virology has advantages and disadvantages, but all of them are very useful for arbovirus diagnostic depending on the kind of sample, storage conditions, objective of the investigation etc.

Please, update sentences from line 58 to 68 according to the comments above.

Line 90 to 96 – I know these samples were collected between 2015 and 2016, but information regarding sampling is limited. The volume and storage conditions of each type of sample was not clearly described. What was the volume collected of urine and saliva samples? How these samples were collected? Were they diluted in viral transport media or kept in natura? Were they kept in the refrigerator until tested? If that is the case, for how long before RNA extraction? Were these samples collected between 2015 and 2016, but tested recently? I would guess that is the case because the RT-PCR protocol used was published in 2020. If I’m assuming right it would be interesting to discuss the longevity of RNA observed in these ZIKV samples. If all that information is available please add to the material and methods section. If not, mention the absence of that information, and then discuss later its impact on the analysis.

Line 93 a 96 – Please, describe here also when the second sample was collected. For instance, second samples were collected from XX to YY days after the first sampling or onset of symptoms. Also, add “(table 1)” at the end of line 96.

Line 104 to 116 – Please, describe here the different targets of ZIKV and CHIKV used for both methods A and B designed for each virus.

Line 116 – Is there any reason to use Cq value instead of Ct value, which is more commonly used? If so, please comment here?

Table 1 – Replace method A and B by the target gene used in each method. Also, the official acronyms of Zika and Chikungunya viruses are ZIKV and CHIKV. Update accordingly.

Figure 1A – This graph shows the probability of detection according to the days of symptoms. Only a few patients had sequential samples, and all of these patients had only one extra sample. For a better evaluation it would be recommended to test more sequential samples. The detection along time is clearly not the main objective of this study and the presentation of these data needs more consideration. If keeping this analysis, its limitation needs to be addressed in the discussion section. Also, in Figure 1A, replace “Days after symptoms” by “Days since onset of symptoms”  

Figure 1B – This graph is clearly showing the Cq difference among sample types. However, its legend states the following: (B), The significance of the difference between the curves in (A) is indicated by the p-value, based on the difference of the Cq values (Table 1). The p-values in bold are considered as significant (<0.05). The way that Figure 1B is presented it is not very clear how the difference of probability of detection is related to the Cq difference. To calculate the probability of detection is necessary the number of samples that tested positive (Cq<38) over the number of samples that were tested in each group, right? The differences in Cq values among the positive samples would not impact the results, right? I see two main points, the first one is that urine and saliva are more likely to be positive than sera, and the second one is that urine and saliva have lower Cq values when compared to sera. That said, I would suggest to rephrase the sentence: “The significance of the difference between the curves in (A) is indicated by the p-value, based on the difference of the Cq values (Table 1).“

Regarding the drawing of a red and blue boxing gloves, I would recommend checking with the journal instructions to authors to see if this sort of presentation is desirable by the journal.

Line 192 to 204 – About the conclusions, considering that information about sample collection and storage is not fully described, I would suggest a more conservative conclusion, or maybe even a different title for this manuscript as “Retrospective detection of ZIKV RNA in urine and saliva samples”. The detection of RNA in a particular sample depends on several factors, as listed in a comment above and if instrumental information is not available this limitation of the study must be addressed and discussed.

Author Response

Reply to the reviewers’ comments

Reviewer #4

1-Please Line 35 – Please, include a reference for the first sentence here.

References were added.

Line 35 – I suggest describing the taxonomic acronym ZIKV here, as Zika virus (ZIKV), and then use the acronym ZIKV elsewhere in the manuscript.

This was adapted.

Line 35 – If describing the taxonomic entity genus Flavivirus, it should be in italics: Flavivirus. If only referring to the group of flaviviruses, use no italics and no capital letter: flavivirus.

This was adapted.

Line 35 – This sentence is too long with five lines. I suggest dividing it in two or three sentences or if keeping as one sentence reduce the amount of details to describe ZIKV occurrence worldwide. “Among the emerging pathogens, Zika virus, a single-strand RNA Flavivirus has been responsible of several outbreaks, such as in 2007 in Yap, Federated States of Micronesia, Pacific region, in 2013-2014 in French Polynesia, Cook Islands, Vanuatu, Salomon Islands, 37 Ester Islands and New Caledonia and other Pacific Islands, and in 2014-2016 in Central 38 America, Latin America, the Caribbean, the United States and the Pacific region.” Suggestion: “Among the emerging pathogens, Zika virus (ZIKV), a single-strand RNA flavivirus has been responsible of several outbreaks, such as in 2007 in the Federated States of Micronesia, in 2013-2014 in New Caledonia and other Pacific Islands, and in 2014-2016 in Latin America and the United States.”

The sentence was adapted as suggested.

Line 43 – Please, add a reference after “traveling”

References were added.

Line 43 – Please, from now on use the acronym ZIKV for all “Zika virus”

This was adapted.

“The Zika virus can then also be passed from infected humans to other humans through body fluids…” Suggestion: “ZIKV can also be transmitted through body fluids…”

The sentence was modified as suggested.

Line 46 – Replace “have symptoms” by “have symptoms and clinical signs”. Fever, rash and conjunctivitis are not symptoms. Change that elsewhere in the manuscript.

The sentence was adapted.

Line 54 – Replace “medical” by “clinical”

This modification was integrated.

Line 57 – Replace “further” by “laboratorial”

The sentence was adapted.

Line 59 – Replace “ineffective for the early stage diagnosis, which is essential for the treatment of individual patients and, even more, for the population in terms of public health.” by “less sensitive for the very early stage of infection.”

The sentence was modified based on this comment.

Line 61 – Delete “or pandemic”

This was adapted.

Line 63 – “However, such techniques are ineffective for the early stage diagnosis, which is essential for the treatment of individual patients and, even more, for the population in terms of public health. Indeed, the ability to quickly identify the causative pathogen of an emerging epidemic or pandemic markedly increases the chances of success for any countermeasure to constrain the disease. Moreover, serological techniques present usually a lack of specificity due to the possible cross reactivity with antibodies from infections associated to other arboviruses and, even, other pathogens. Therefore, to overcome these crucial issues, molecular approaches targeting nucleic acids of interest, such as the reverse transcription real-time polymerase chain reaction (RT-qPCR), are generally privileged.”

Laboratorial methods designed to detect antibodies are the most valuable diagnostic tools for arbovirus diagnosis. There are around 500 arboviruses officially reported to the International Committee on Taxonomy of Viruses worldwide and the large majority of them is kept in enzootic cycles of transmission, which humans are accidental hosts for not mounting viremia high and long enough to act as a source of infection to arthropods. Very few exceptions including zika, dengue, chikungunya, and yellow fever, partially, have humans as amplifying hosts and only for these arboviruses the use of direct laboratorial diagnosis as virus isolation and RT-PCR is likely to be successful. It is noteworthy that the direct diagnostic is more sensitive only for a very short period of time and the success of the diagnostic is dependent on the sample storage conditions and viremia duration, while antibodies that are proteins, are more resistant than RNA and can be detected for a longer time after exposure. Indeed, the cross-reactivity observed among flaviviruses influences serological diagnostic results and the interpretation of serological tests is complex requiring careful evaluation. However, there are serological methods with high specificity as the plaque reduction neutralization test (PRNT), which is able to differentiate flavivirus infections in convalescent serum samples. Two or more flaviviruses can be distinct from each other by quantitative serological criteria. Thus, each laboratorial method used in virology has advantages and disadvantages, but all of them are very useful for arbovirus diagnostic depending on the kind of sample, storage conditions, objective of the investigation etc.

Please, update sentences from line 58 to 68 according to the comments above.

This point was adapted based on the comments above.

Line 90 to 96 – I know these samples were collected between 2015 and 2016, but information regarding sampling is limited. The volume and storage conditions of each type of sample was not clearly described. What was the volume collected of urine and saliva samples? How these samples were collected? Were they diluted in viral transport media or kept in natura? Were they kept in the refrigerator until tested? If that is the case, for how long before RNA extraction? Were these samples collected between 2015 and 2016, but tested recently? I would guess that is the case because the RT-PCR protocol used was published in 2020. If I’m assuming right it would be interesting to discuss the longevity of RNA observed in these ZIKV samples. If all that information is available please add to the material and methods section. If not, mention the absence of that information, and then discuss later its impact on the analysis.

Additional information (volume and storage) was added (section 2.1).

Samples were collected in 2015-2016. RNA extraction was performed in 2016-2017. The real-time PCR analysis was performed in 2017-2018. This information was added in the manuscript (sections 2.1-2.2). On this basis, a discussion about the longevity of RNA is not relevant. 

Line 93 a 96 – Please, describe here also when the second sample was collected. For instance, second samples were collected from XX to YY days after the first sampling or onset of symptoms.

Table S1, with such information, was added.

Also, add “(table 1)” at the end of line 96.

This was added.

Line 104 to 116 – Please, describe here the different targets of ZIKV and CHIKV used for both methods A and B designed for each virus.

The quadruplex real-time PCR was not developed in this study. Such information are detailed in Broeders et al., 2020. This point was clarified in section 2.2 and section 3. Moreover, the names of the targeted genes by each methods were added in Table S2.

Line 116 – Is there any reason to use Cq value instead of Ct value, which is more commonly used? If so, please comment here?

Both terms (Cq and Ct) are used. We decided to use the term Cq as introduce in the MIQE guidelines, a reference peer-reviewed publication for real-time PCR (Bustin et al. 2009).

Table 1 – Replace method A and B by the target gene used in each method.

To be consistent with Broeders et al. 2020 (describing the developed of these methods), we prefer to keep the original names of the methods. The names of the targeted genes for each method were added in Table S2.

Also, the official acronyms of Zika and Chikungunya viruses are ZIKV and CHIKV. Update accordingly

This was adapted.

Figure 1A – This graph shows the probability of detection according to the days of symptoms. Only a few patients had sequential samples, and all of these patients had only one extra sample. For a better evaluation it would be recommended to test more sequential samples. The detection along time is clearly not the main objective of this study and the presentation of these data needs more consideration. If keeping this analysis, its limitation needs to be addressed in the discussion section.

As suggested, the importance to use a larger number of samples and patients to support our observations was added in the last paragraph of the section results and discussion.

Also, in Figure 1A, replace “Days after symptoms” by “Days since onset of symptoms” 

This was adapted.

Figure 1B – This graph is clearly showing the Cq difference among sample types. However, its legend states the following: (B), The significance of the difference between the curves in (A) is indicated by the p-value, based on the difference of the Cq values (Table 1). The p-values in bold are considered as significant (<0.05). The way that Figure 1B is presented it is not very clear how the difference of probability of detection is related to the Cq difference. To calculate the probability of detection is necessary the number of samples that tested positive (Cq<38) over the number of samples that were tested in each group, right? The differences in Cq values among the positive samples would not impact the results, right? I see two main points, the first one is that urine and saliva are more likely to be positive than sera, and the second one is that urine and saliva have lower Cq values when compared to sera. That said, I would suggest to rephrase the sentence: “The significance of the difference between the curves in (A) is indicated by the p-value, based on the difference of the Cq values (Table 1).“

We thank the reviewer for this comment. Indeed, these values did not reflect the Cq value difference and have been left out.

Regarding the drawing of a red and blue boxing gloves, I would recommend checking with the journal instructions to authors to see if this sort of presentation is desirable by the journal.

The drawing was removed.

Line 192 to 204 – About the conclusions, considering that information about sample collection and storage is not fully described, I would suggest a more conservative conclusion, or maybe even a different title for this manuscript as “Retrospective detection of ZIKV RNA in urine and saliva samples”. The detection of RNA in a particular sample depends on several factors, as listed in a comment above and if instrumental information is not available this limitation of the study must be addressed and discussed.

Given additional information added in sections 2.1 and 2.2, a discussion about the longevity of RNA is not relevant. 

Round 2

Reviewer 2 Report

Dear author

Please change table 1 caption as the title and explain it in the main text. Abbreviations of the terms used should be footnoted. Nothing much had changed in the table.

Explanations of the table 1 is not satisfactory, still confusing

Table 1 of the supplementary is not what I meant by my comment. A summery table of the signs and symptoms is still needed. 

Discussion and conclusion has not changed and authors reply is not satisfactory.

Author Response

Reply to the comments

Reviewer #2

Please change table 1 caption as the title and explain it in the main text. Abbreviations of the terms used should be footnoted. Nothing much had changed in the table. Explanations of the table 1 is not satisfactory, still confusing

The explanation of the table was moved below the table. In addition, if necessary, information was added in the text (sections 2.1 and 2.2).

Table 1 of the supplementary is not what I meant by my comment. A summery table of the signs and symptoms is still needed.

The table S1 is a summary of all symptoms and clinical signs reported for all patients samples. The title was adapted to avoid any misunderstanding. If this table is not corresponding to the demand of reviewer 2, is it possible to receive clear instruction for potential modification.  

Discussion and conclusion has not changed and authors reply is not satisfactory.

The two previous comments from reviewer 2 were the following:

1/ Please rewrite your discussion based on the new suggested results

2/ Conclusion should be the main contributions of your work and avoid referencing in this section

All the demands from reviewer 2 were addressed in the results and discussion and in the conclusion. In addition, to stress the impact of these results, the following adaptations were made:

  • Adaptations in the results and discussion for “Please describe the patient’s characteristics more thoroughly, a table would be great to introduce the symptoms and why you chose them”

The main symptoms and clinical signs from patients are clearly mentioned “Serum, urine and saliva samples were collected in 2015-2016 in New Caledonia and French Guiana from 53 patients presenting symptoms and clinical signs associated with arbovirus infections, including mainly fever, myalgia, arthralgia, cephalgia, nausea, retro-orbital pain and skin rash (Table S1)”. All symptoms and clinical signs (including the less frequent ones) are also available in Table S1.

  • Adaptations in the results and discussion for “Please provide sensitivity and specificity as better figures to evaluate each test”

The 4-plex real-time PCR, developed by Broeders et al. (2020) (as specified through the manuscript), were previously assessed as specific and sensitive. This point is also now added in the first paragraph of the results and discussion: “To this end, a 4-plex RT-qPCR targeting both CHIKV, via the CHIKV (a) and CHIKV (b) methods, and ZIKV, via the ZIKV (a) and ZIKV (b) methods, previously developed and assessed as specific and sensitive by Broeders et al (2020), was applied”.

  • Adaptations in the results and discussion as well as in the conclusion for “comparing should be depicted using bars and p values; we cannot see which test is superior based on your figures”

In addition to the adaptations previously done in Figure 1, the superior tests, being the use of urine as well as the use of both urine and saliva, are now clearly mentioned in the results and discussion (“On this basis, the use of urine samples to diagnose ZIKV, representing the most appropriate individual test, is thus strongly advised”; “The use of both the urine and saliva samples, representing the most appropriate combined test, is thus strongly advised”) and in the conclusion (“Indeed, the use of urine samples as well as the use of both urine and saliva samples were highlighted as being the most appropriated tests in this study”).

  • Adaptations in the conclusion for “Conclusion should be the main contributions of your work and avoid referencing in this section”

The main contribution of the present study are clearly mentioned (“Based on the present study, the use of one or a combination of non-invasive samples instead of serum samples is highly recommended for the detection of viral RNA from ZIKV at early stage. Indeed, the use of urine samples as well as the use of both urine and saliva samples were highlighted as being the most appropriated tests in this study”). In addition, references were removed.

Reviewer 4 Report

Please, include in the title more information about the subset of samples tested. The way it is sounds like a large worldwide study.

Suggestion:

Non-invasive versus invasive samples for Zika virus surveillance: a comparative study in New Caledonia and French Guiana in 2015-2016

There are few studies showing that ZIKV RNA lasts for a longer time after onset of infection in urine compared to blood. Please, stress that information in the introduction with some references.

Line 67 - "However, such techniques are less sensitive for the early stage diagnosis, which is essential for the treatment of individual patients and, even more, for the population in terms of public health. Indeed, the ability to quickly identify the causative pathogen of an emerging epidemic or pandemic markedly increases the chances of success for any countermeasure to constrain the disease. Moreover, serological techniques, with exceptions (i.e., plaque reduction neutralization test), present usually a lack of specificity due to the possible cross reactivity with antibodies from infections associated to other arboviruses and, even, other pathogens. Therefore, to overcome these crucial issues, molecular approaches targeting nucleic acids of interest, such as the reverse transcription real-time polymerase chain reaction (RT-qPCR), are generally privileged

Replace by: 

"However, such techniques are less sensitive for the very first days after infection, which is essential to quickly identify the causative pathogen of an emerging epidemic, and increases the chances of success for any countermeasure to constrain the disease."

There is no need to downplay antibody investigation to justify the use of RT-PCR in the present study. As said before, serology is highly valuable laboratorial method for arbovirus diagnosis and the way the authors present mislead the readers. The authors should focus on the method used and avoid comparing the two approaches. Besides, since the authors were comparing samples as urine, there would be no point in using serology is such kind of sample to evaluate ZIKV and CHIKV infections. 

Table 2 - "The Cq value difference is based on the lowest RT-qPCR signal observed for each compared sample that were both presenting a positive signal (Cq value ≤ 38). The p-value in bold is considered as significant (<0.05)."

It might not make much difference, but prefer using average of Cq instead of lowest values.

Author Response

Reply to the comments

Reviewer #4

Please, include in the title more information about the subset of samples tested. The way it is sounds like a large worldwide study.

Suggestion:

Non-invasive versus invasive samples for Zika virus surveillance: a comparative study in New Caledonia and French Guiana in 2015-2016

The title was adapted as suggested.

There are few studies showing that ZIKV RNA lasts for a longer time after onset of infection in urine compared to blood. Please, stress that information in the introduction with some references.

This point was indicated in the introduction “. In addition, the use of urine samples offered a broader detection window compared to the use of saliva and serum samples”. To stress it, additional references were added.

Line 67 - "However, such techniques are less sensitive for the early stage diagnosis, which is essential for the treatment of individual patients and, even more, for the population in terms of public health. Indeed, the ability to quickly identify the causative pathogen of an emerging epidemic or pandemic markedly increases the chances of success for any countermeasure to constrain the disease. Moreover, serological techniques, with exceptions (i.e., plaque reduction neutralization test), present usually a lack of specificity due to the possible cross reactivity with antibodies from infections associated to other arboviruses and, even, other pathogens. Therefore, to overcome these crucial issues, molecular approaches targeting nucleic acids of interest, such as the reverse transcription real-time polymerase chain reaction (RT-qPCR), are generally privileged

Replace by:

"However, such techniques are less sensitive for the very first days after infection, which is essential to quickly identify the causative pathogen of an emerging epidemic, and increases the chances of success for any countermeasure to constrain the disease."

There is no need to downplay antibody investigation to justify the use of RT-PCR in the present study. As said before, serology is highly valuable laboratorial method for arbovirus diagnosis and the way the authors present mislead the readers. The authors should focus on the method used and avoid comparing the two approaches. Besides, since the authors were comparing samples as urine, there would be no point in using serology is such kind of sample to evaluate ZIKV and CHIKV infections.

These sequences were adapted as suggested.

Table 2 - "The Cq value difference is based on the lowest RT-qPCR signal observed for each compared sample that were both presenting a positive signal (Cq value ≤ 38). The p-value in bold is considered as significant (<0.05)."

It might not make much difference, but prefer using average of Cq instead of lowest values.

For each kind of samples (urine, saliva and serum), two RT-qPCR methods targeting Zika virus, being ZIKV (a) and ZIKV (b), were tested in duplicate. For each method, the average Cq was indicated in Table 1. The average Cq values from the ZIKV (a) method were in general lower than the ones from the ZIK (b) method. Moreover, the ZIK (a) method presented more positive signal than the ZIK (b) method. As a patient was considered as infected when at least one of these two ZIKV methods was positive, the Cq value from the ZIK (a) method indicated in Table 1 was taken into account to perform the analysis in Table 2. This information was specified in the sections 2.2 and 2.3.2, as well as in Table 2.
